# Detection of SARS-CoV-2 Delta Variant (B.1.617.2) in Domestic Dogs and Zoo Tigers in England and Jersey during 2021

**DOI:** 10.3390/v16040617

**Published:** 2024-04-16

**Authors:** Amanda H. Seekings, Rebecca Shipley, Alexander M. P. Byrne, Shweta Shukla, Megan Golding, Joan Amaya-Cuesta, Hooman Goharriz, Ana Gómez Vitores, Fabian Z. X. Lean, Joe James, Alejandro Núñez, Alistair Breed, Andrew Frost, Jörg Balzer, Ian H. Brown, Sharon M. Brookes, Lorraine M. McElhinney

**Affiliations:** 1Department of Virology, Animal and Plant Health Agency-Weybridge, Woodham Lane, New Haw, Addlestone, Surrey KT15 3NB, UK; 2National Reference Laboratory for SARS-CoV-2 in Animals, Animal and Plant Health Agency-Weybridge, Woodham Lane, New Haw, Addlestone, Surrey KT15 3NB, UK; 3Worldwide Influenza Centre, The Francis Crick Institute, Midland Road, London NW1 1AT, UK; 4Department of Pathology and Animal Sciences, Animal and Plant Health Agency-Weybridge, Woodham Lane, New Haw, Addlestone, Surrey KT15 3NB, UK; 5Government of Jersey, Infrastructure Housing and Environment, Howard Davis Farm, La Route de la Trinité, Trinity, Jersey JE3 5JP, UK; 6One Health, Animal Health and Welfare Advice Team, Animal and Plant Health Agency, Nobel House, 17 Smith Square, London SW1P 3JR, UK; 7Vet Med Labor GmbH, Division of IDEXX Laboratories, Humboldtstraße 2, 70806 Kornwestheim, Germany

**Keywords:** SARS-CoV-2, Delta variant, reverse zoonosis, dog, cat, tiger

## Abstract

Reverse zoonotic transmission events of severe acute respiratory syndrome coronavirus 2 (SARS-CoV-2) have been described since the start of the pandemic, and the World Organisation for Animal Health (WOAH) designated the detection of SARS-CoV-2 in animals a reportable disease. Eighteen domestic and zoo animals in Great Britain and Jersey were tested by APHA for SARS-CoV-2 during 2020–2023. One domestic cat (*Felis catus*), three domestic dogs (*Canis lupus familiaris*), and three Amur tigers (*Panthera tigris altaica*) from a zoo were confirmed positive during 2020–2021 and reported to the WOAH. All seven positive animals were linked with known SARS-CoV-2 positive human contacts. Characterisation of the SARS-CoV-2 variants by genome sequencing indicated that the cat was infected with an early SARS-CoV-2 lineage. The three dogs and three tigers were infected with the SARS-CoV-2 Delta variant of concern (B.1.617.2). The role of non-human species in the onward transmission and emergence of new variants of SARS-CoV-2 remain poorly defined. Continued surveillance of SARS-CoV-2 in relevant domestic and captive animal species with high levels of human contact is important to monitor transmission at the human−animal interface and to assess their role as potential animal reservoirs.

## 1. Introduction

Severe acute respiratory syndrome coronavirus 2 (SARS-CoV-2), the causative agent of coronavirus disease 2019 (COVID-19), was first reported in humans in Wuhan, China in late 2019 [1]. Widespread transmission and circulation of SARS-CoV-2 in humans resulted in COVID-19 being declared a pandemic by the World Health Organisation (WHO) in March 2020 [2]. As of 28 January 2024, approximately 774 million human cases have been reported with more than seven million deaths worldwide [3]. The extended circulation of SARS-CoV-2 has led to the accumulation of viral genetic mutations resulting in the emergence of numerous viral lineages, some of which have been classified as variants of concern (VOCs). Since the early characterised SARS-CoV-2 strains from 2019 and early 2020, the WHO has globally defined five VOCs, namely, Alpha (PANGO [4] lineage B.1.1.7), Beta (B.1.351), Gamma (P.1), Delta (B.1.617.2) and its descendant AY sublineages, and Omicron (B.1.1.529) and its descendant BA sublineages. Emerging variants are continually assessed for their concerning properties such as their impact on transmission and clinical disease outcomes in humans [5].

The Alpha variant originated in the United Kingdom (UK) and spread globally. This was retrospectively determined to have emerged in September 2020 due to multiple mutations in the spike protein [6]. The Beta and Gamma variants rapidly emerged in South Africa [7] and Brazil [8], respectively, with increased incidence observed by December 2020. The Delta variant was first detected in India in December 2020 [9] and was designated a VOC in the UK in May 2021. Human infection with the Delta variant was associated with more severe disease compared to previous VOCs with higher replication and transmission potential [10,11]. Continued extensive transmission among the human population resulted in several descendant sublineages of the Delta variant designated with the ‘AY’ prefix. In particular, the Delta variant sublineage AY.4.2 emerged in July 2021 in England and detections increased steadily until December 2021 when the Omicron VOC emerged and became more prevalent [12].

The origins of SARS-CoV-2 remain unknown, although bats are implicated as the likely original host, with evidence suggesting an unidentified intermediate animal host was involved prior to transmission to humans [13,14,15]. Significant gaps remain in our understanding of the role non-human species may play in the transmission of SARS-CoV-2, and the ability of these animals to act as reservoirs and/or amplifying hosts. As such, the World Organisation for Animal Health (WOAH) designated the detection of SARS-CoV-2 in animals an internationally reportable disease and a regulatory framework in the UK outlined reporting obligations of SARS-CoV-2 in mammals from February 2021 [16]. As of 24 October 2023, a total of 775 animal cases have been reported to WOAH [17] from 36 countries in the Americas, Africa, Asia, and Europe involving 29 different animal species including felids (domestic cats, tigers, lions and snow leopards), canids (domestic dogs and red foxes), mustelids (mink and domestic ferrets), rodents (hamsters), and cervids (white-tailed deer). Many of these detections are a result of reverse zoonotic transmission events. SARS-CoV-2 transmission from humans to companion animal species has been most notable in domestic cats, ferrets, and hamster species where there are frequent human interactions; however, the risk of spillback infection from these species to humans remains low [18]. The potential for animal reservoirs of SARS-CoV-2 to become established is the focus of continued surveillance and risk assessment in key animal species [18,19]. In particular, SARS-CoV-2 infections have been reported in farmed American mink in Denmark, the Netherlands, Greece, USA, and Sweden [20] and represent the first animal species where sustained intraspecies transmission was detected. Extended SARS-CoV-2 transmission in farmed mink resulted in the generation of novel mink-specific variants with evidence of spillback to humans [21]. In farmed white-tailed deer in North America, multiple SARS-CoV-2 reverse zoonotic events have been reported with subsequent intraspecies transmission [22,23]. Continued SARS-CoV-2 transmission among deer resulted in accelerated viral evolution distinct from the evolutionary trajectories of SARS-CoV-2 viruses circulating in the human population [22,24,25], highlighting the potential for establishing an animal reservoir for SARS-CoV-2.

The National Reference Laboratory (NRL) for SARS-CoV-2 in animals in Great Britain (GB) at the Animal and Plant Health Agency (APHA) provides confirmatory testing of domestic felids, canids, and species of the *Mustelinae* family (including ferrets and mink), and provides primary testing of non-domestic species of large felid, non-human primate, and any *Mustelinae* kept in captivity with suspicion of SARS-CoV-2 infection, according to specified case definitions [16]. This case report summary describes the animal submissions received at the APHA for primary or confirmatory SARS-CoV-2 testing and details the seven positive animal cases detected and reported to WOAH during 2020–2022. The findings re-iterate the need for continued surveillance of animal SARS-CoV-2 cases, further contributing to our understanding of the epidemiology, control measures, and risk assessment at the human−animal interface.

## 2. Materials and Methods

### 2.1. Sample Collection and Processing

Conjunctival, oral (pharyngeal, mouth, throat, saliva, sputum), nasal or rectal swabs were taken from live animals or carcasses post-mortem and suspended in 1 mL phosphate-buffered saline (PBS) or Dulbecco’s modified Eagle’s media (DMEM, Gibco, Grand Island, NY, USA) supplemented with 1% (*v*/*v*) Penicillin and 1% (*v*/*v*) Streptomycin. Tissues or additional swabs collected from organs harvested at post-mortem are indicated (Table 1) and faeces samples were either suspended in DMEM, added directly to 1 ml TRIzol Reagent (Thermofisher Scientific, Loughborough, UK) or 1 mL lysis buffer (MagMAX Total Nucleic Acid Isolation Kit, Thermofisher Scientific) for homogenisation and downstream RNA extraction. Bronchoalveolar lavage fluid (BALF) was taken and added directly to lysis buffer for RNA extraction. The serum samples obtained were heated at 56 °C for 30 min to inactivate antibody complement and then stored at 4 °C or −20 °C prior to testing. Samples used for histopathological and in situ detection of SARS-CoV-2 were fixed in neutral-buffered 10% formalin fixative solution and routinely processed as described [26].

### 2.2. RNA Extraction

Total RNA was extracted from all samples using either the QIAmp Viral RNA Mini Kit (Qiagen, Manchester, UK), MagMAX Total Nucleic Acid Isolation Kit using the Kingfisher Flex System (Thermofisher Scientific, Loughborough, UK) according to the manufacturer’s instructions or by adding TRIzol Reagent (Thermofisher Scientific, Loughborough, UK) followed by organic solvent extraction using chloroform and resuspended in RNAse free water.

**Table 1 viruses-16-00617-t001:** Summary of submissions to the National Reference Laboratory for SARS-CoV-2 in animals in GB.

Submission #	Sample Collection Date ^1^	Species, Breed	Age	Sex	Sample Type	E gene RRT-PCR (Cq)	RdRp RRT-PCR (Cq)	Virus Neutralisation Titre (IC_50_)	Interpretation
1	22 April 2020	Cat, Ragdoll	4 months	Female	Lung tissue (formalin fixed paraffin embedded)	No Cq	No Cq	n/a	Negative
2	15 May 2020	Cat, Siamese	6 years	Female	Oropharyngeal swab	32.00	34.62	n/a	Positive [27]
	10 July 2020				Oropharyngeal swab	No Cq	No Cq	n/a	
					Rectal swab	No Cq	No Cq	n/a	
					Serum	n/a	n/a	128	
3	15 January 2021	Dog, Pug	Unknown	Unknown	Saliva swab	No Cq	No Cq	n/a	Negative
4	29 January 2021	Cat, unknown	Unknown	Unknown	Nasal, mouth, throat, rectal swabs	No Cq	nt	n/a	Negative
5	5 February 2021	Cat, Sphynx	10 years, 7 months	Female	Throat, nasal, conjunctival and rectal swab, 28 additional tissues collected post-mortem	No Cq	No Cq	n/a	Negative
					Serum	n/a	n/a	Negative	
6	21 April 2021	Camel, Bactrian	Unknown	Unknown	Lung tissue	No Cq	No Cq	n/a	Negative
7	9 July 2021	Dog, poodle × shih tzu cross	8 years	Female	Conjunctival and oropharyngeal swab pooled	33.42	38.65	n/a	Positive
8	7 September 2021	Gaur Indian Bison	Unknown	Unknown	Lung tissue	No Cq	No Cq	n/a	Negative
9	29 October 2021	Dog, Labrador	10 years	Male	Pharyngeal swab Rectal swab Conjunctival swab	24.29 No Cq No Cq	30.32 No Cq No Cq	n/a	Positive
10	3 December 2021 9 December 2021	Tiger, Amur	12 years	Male	Sputum swab	29.08	34.59	n/a	Positive
					Virus transport medium wash from sputum swab	27.60	nt		
					Oral swab	No Cq	No Cq		
					Nasal swab	No Cq	No Cq		
11	9 December 2021	Tiger, Amur	13 years	Female	Nasal swab	35.83	No Cq	n/a	Positive
12	9 December 2021	Tiger, Amur	11 years	Female	Oral swab Nasal swab Faeces	36.98 31.46 36.99	No Cq No Cq No Cq	n/a	Positive
13	9 December 2021	Leopard, Amur	12 years	Male	Oral swab Nasal swab Faeces	No Cq No Cq No Cq	No Cq No Cq No Cq	n/a	Negative
14	9 December 2021	Cat, Burmese	Unknown	Unknown	Oral swab	No Cq	No Cq	n/a	Negative
15	6 December 2021	Dog, Labrador	4 years 10 months	Male	Pharyngeal swab RNA	31.46	nt	n/a	Positive
	13 December 2021				Nasal swab Throat swab BALF Blood Brain tissue Additional 25 tissues	32.75 34.75 30.50 33.44 31.71 No Cq	37.01 39.85 35.73 No Cq 35.67 No Cq		
16	17 December 2021	Asian palm civet cat	Adult	Female	Oropharyngeal, conjunctival, nasal, vagina, rectal swab, BALF, additional 25 tissues	No Cq	No Cq	n/a	Negative
17	10 February 2022	Cat, Short hair British	8 months 4 weeks	Male	Left and right caudal and cranial lung, nasal turbinates, pooled organs (liver, kidney, spleen, thymus) collected at post-mortem	No Cq	No Cq	n/a	Negative
18	3 November 2023	Gorilla	19 years	Male	Nasal swab	No Cq	No Cq	n/a	Negative

n/a, not applicable; nt, not tested; ^1^ sample collection date given where available, otherwise date received at APHA provided.

### 2.3. Real-Time Reverse-Transcription Polymerase Chain Reaction (RRT-PCR)

Viral RNA (vRNA) was detected using real-time reverse-transcription PCR (RRT-PCR) assays at APHA targeting the SARS-CoV-2 envelope (E) and RNA-dependent RNA polymerase (RdRp) genes using primers and probes described previously [28] in a duplex reaction with beta actin primers and probe as described previously [29]. A 25 μL reaction was prepared using the AgPath-ID™ One-Step RT-PCR Reagent Kit (Applied Biosystems, Warrington, UK). Seven microliters of extracted RNA was added to each reaction mix and cycling conditions were as follows: 45 °C for 10 min for reverse transcription, followed by 95 °C for 10 min and then 45 cycles of 95 °C for 15 s and 60 °C for 30 s using the AriaMx Real-Time PCR System (Agilent Technologies, Santa Clara, CA, USA). Fluorescence data were gathered at the end of each 60 °C step. Quantity of vRNA was expressed as quantification cycle (Cq) values. A positive result is determined by a Cq value corresponding to Cq ≤ 37.00. The RRT-PCR assays undertaken at IDEXX for Submission 7 included IDEXX in house SARS-CoV-2 RRT-PCR, CDC N1, N2 and N3 PCRs targeting the nucleocapsid phosphoprotein (N) as described previously [30].

### 2.4. Whole-Genome Sequencing and Phylogenetic Analysis

Positive samples identified by SARS-CoV-2 specific RRT-PCRs were selected for direct whole-genome sequencing (WGS). Firstly, vRNA extracted as described in Section 2.2 above was used to generate double-stranded cDNA using the NEBNext^®^ ARTIC SARS-CoV-2 RT-PCR Module V1 (New England Biolabs, Ipswich, MA, USA). Library preparation was performed using the Nextera DNA Library Prep kit (Illumina, Cambridge, MA, USA), with 1 ng of double-stranded cDNA product then sequenced using the MiSeq System (Illumina) following the manufacturer’s instructions. Paired-end Illumina reads were assembled using a custom reference guided alignment script (https://github.com/APHA-VGBR/WGS_Pipelines/blob/7f73c31629f483994b8aa366e157028abf69f824/RefGuidedAlignment_Public.sh, accessed on 12 December 2023) using SARS-CoV-2/human/China/WIV04/2020 (GenBank Accession No: MN996528) as the reference sequence. The final consensus was generated using iVar with a minimum depth of one sequence with a 50% minimum frequency rule for degenerate base calling. The quality threshold was a Phred score of 20. All genome consensus sequences obtained have been made available through the Global Initiative on Sharing Avian Influenza Data (GISAID) EpiCoV platform (https://www.GISAID.org) and GenBank (Table 2). The assembled raw reads have been deposited on the NCBI Sequence Read Archive (SRA) database. Genome sequences obtained were assigned a lineage utilising the Phylogenetic Assignment of Named Global Outbreak Lineages (PANGO) algorithm (https://cov-lineages.org/) [4]. Human SARS-CoV-2 genome sequences representative of the variants of concern in the UK from May 2020 to December 2021 were downloaded from the GISAID EpiCoV platform and randomly sub-sampled using Augur [31] before being aligned with the genome sequences obtained from animals using Mafft v7.487 [32]. Multiple sequence alignments followed by nucleotide and amino acid sequence comparisons were carried out using MEGA X. A phylogenetic tree was inferred using the maximum likelihood method in IQ-Tree v2.1.4 [33] with a best-fit model applied using ModelFinder [34] and a phylogeny test of 1000 ultrafast bootstrap replicates [35]. The phylogenetic tree was visualised using FigTree v1.4.2.

### 2.5. SARS-CoV-2 Virus Neutralisation Test (VNT)

The virus neutralisation test (VNT) was undertaken as described previously [36]. Briefly, in a 96-well plate format, two-fold dilutions of the serum sample were made in DMEM supplemented with 2% (*v*/*v*) heat inactivated foetal calf serum (FCS) and 1% (*v*/*v*) Penicillin, Streptomycin, and Nystatin. The SARS-CoV-2 strain hCoV-19/Italy/LAZ-INMI1-isl/2020 (GISAID Accession EPI_ISL_410545) was diluted to 100 tissue culture infectious dose 50% (TCID_50_) units and a volume of 50 µL added to each well. Plates were incubated at 37 °C with 5% CO_2_ for one hour followed by the addition of 4–5 × 10^5^ cells/mL of the Vero E6 cell suspension. The plates were further incubated for a maximum of five days at 37 °C with 5% CO_2_ and visualised daily for cytopathic effect (CPE) using a microscope. The neutralising titre was calculated using the Spearman−Karber method and displayed as inhibitory concentration 50% (IC_50_).

### 2.6. Virus Isolation

Virus isolation was attempted for SARS-CoV-2 positive clinical samples identified by RRT-PCR. Clinical material suspended in DMEM or PBS was added to a flask of confluent Vero/hSLAM cells [37] supplemented with 2% heat inactivated FCS with 1% (*v*/*v*) Penicillin, Streptomycin, and Nystatin. The cells were incubated for a maximum of seven days at 37 °C with 5% CO_2_ and visualised daily for CPE using a microscope. Where required, a maximum of three passages was attempted.

## 3. Results

### 3.1. SARS-CoV-2 Investigations in Animals

The NRL for SARS-CoV-2 in animals in GB at the APHA received and tested samples from 18 animal cases for SARS-CoV-2 primary or confirmatory investigations since 2020 (Table 1). In total, seven animals were confirmed as SARS-CoV-2 positive: a domestic cat (*Felis catus*), three domestic dogs (*Canis lupus familiaris*), and three Amur tigers (*Panthera tigris altaica*) and were reported to the WOAH. The first confirmed SARS-CoV-2 positive animal case (Submission 2) was a six-year-old female domestic cat (Siamese) sampled 15 May 2020 in GB as previously described [27,38]. The cat displayed respiratory signs and was initially diagnosed by a private veterinary surgeon (PVS) with feline herpes virus. However, as the pet owners were SARS-CoV-2 positive, an oropharyngeal swab sample collected from the cat was also tested for SARS-CoV-2 and tested positive by RRT-PCR [27]. The original oropharyngeal swab sample collected from the cat was referred to APHA and independently confirmed as positive by SARS-CoV-2 RRT-PCR. The cat recovered from the infection and eight weeks later further oropharyngeal and rectal swabs, and a serum sample, was submitted to APHA for SARS-CoV-2 testing. The swab samples were negative for SARS-CoV-2 but VNT on the serum sample detected SARS-CoV-2 neutralising antibodies (titre 128 IC_50_) providing further evidence for SARS-CoV-2 infection in the cat. The three domestic dogs and three zoo tigers confirmed as SARS-CoV-2 positive are described further in this case report.

### 3.2. Submission 7: Dog from Jersey

An eight-year-old female domestic dog (poodle × shih tzu cross-breed) was admitted to a PVS in Jersey in a collapsed state with signs of severe respiratory distress on 9 July 2021. It was unresponsive to treatment and was euthanised the same evening. Blood obtained from the animal prior to euthanasia demonstrated leukopenia. The veterinary history detailed that the dog had been diagnosed with a heart condition in 2020 and was prescribed medication to treat this. The dog had also recently undergone orthopaedic surgery. Prior to admittance, the dog had been staying with a person who was isolating due to COVID-19. Swabs (one oropharyngeal and one conjunctival) were collected after euthanasia and sent to IDEXX for primary testing. Upon receipt at IDEXX, vRNA was extracted from the pooled swabs and tested using an in-house PCR to detect the SARS-CoV-2 nucleocapsid phosphoprotein (N) gene. The positive PCR result (Cq 34.60) returned from the swabs was then confirmed using three further PCR assays targeting different regions of the N gene (CDC N1: Cq 32.80, N2: Cq 32.30, and N3: Cq 32.80), all of which gave positive results. On receipt of the positive result, the PVS notified the Government of Jersey veterinary section and as the dog originated from Jersey, a Crown Dependency, the original pooled swab eluate was transferred to APHA for confirmatory testing. The vRNA extracted from the pooled swab eluate was tested by SARS-CoV-2 RRT-PCRs and confirmed as positive (Table 1) by the NRL. Virus isolation was attempted on the RRT-PCR positive swab eluate but was unsuccessful. There was no further material available to test or other pets in the household that could have been sampled and it was therefore not possible to re-attempt viral isolation. The positive RRT-PCR result obtained from the swabs was sufficient to suggest the dog was infected with SARS-CoV-2 but there is insufficient evidence to demonstrate if the dog was clinically affected by the virus as the dog had other significant underlying health conditions.

### 3.3. Submission 9: Dog from England

A ten-year-old neutered male domestic dog (Labrador retriever) was presented to a PVS with sudden onset of rapid breathing, inappetence, and raised temperature (40.8 °C) on 29 October 2021. Clinical investigations revealed the dog also had mild lymphopaenia, anaemia, and thrombocytopaenia with moderate pleural effusion. The dog underwent a thoracotomy five weeks prior and recovered without complications. Both owners of the dog residing in the same household had COVID-19 at the time with onset of symptoms from 26 October 2021. A pharyngeal swab was collected from the dog and a positive SARS-CoV-2 result was reported by the PVS. The pharyngeal swab and associated media were referred to APHA for confirmatory testing in addition to rectal and conjunctival swabs from the dog. SARS-CoV-2 vRNA was detected in the pharyngeal swab by RRT-PCR (Table 1) and the virus was isolated. No vRNA was detected from the rectal or conjunctival swabs. The dog received treatment with anti-inflammatories, antibiotics, and an anthelminthic. The dog and owners recovered from the SARS-CoV-2 infection.

### 3.4. Submissions 10, 11, and 12: Tigers from England

Two captive Amur tigers housed in the same enclosure in a wildlife park in England began to show clinical signs of coughing, diarrhoea, reduced appetite, and lethargy on 30 November 2021. A sputum sample from one of the tigers, a 12-year-old male, was collected on 3 December 2021 and a swab taken from the sputum sample tested positive for SARS-CoV-2 by RRT-PCR at a private laboratory. The sample was referred to APHA for confirmatory testing. Oral and nasal swab samples were collected on 9 December 2023 from the same 12-year-old male Amur tiger (Submission 10), a 13-year-old female Amur tiger (Submission 11) that also had similar clinical signs, and a third Amur tiger, 11-year-old female (Submission 12), that was not reported to have any clinical signs but were all housed in the same enclosure. Faecal samples were also collected from the 11-year-old female Amur tiger (Submission 12). At the time of sampling on 9 December 2021, the two tigers (Submissions 10 and 11) were reported to be recovering. The tigers were not on public display and interaction with workers was restricted to minimise the risk of SARS-CoV-2 transmission. There were no known human contacts with confirmed COVID-19 cases at the time of sampling; however, one worker was COVID-19 positive three months prior. The vRNA extracted from the sputum swab sample and the virus transport medium wash from the sputum sample collected on 3 December 2021 were positive for SARS-CoV-2 E-gene with Cq values of 29.08 and Cq 27.60, respectively (Submission 10; Table 1). The oral and nasal swab samples collected from the same tiger six days later, on 9 December 2021, were negative (no Cq) by SARS-CoV-2 RRT-PCRs. The nasal swab sample collected from the second tiger (Submission 11) was positive for SARS-CoV-2 vRNA at Cq 35.83. Oral swab, nasal swab, and faeces samples collected from the third tiger (Submission 12) were also confirmed to be positive for SARS-CoV-2 vRNA at Cq 36.98, Cq 31.46, and Cq 36.99, respectively. Virus isolation from the tiger samples after three passages was unsuccessful. An Amur leopard in a different enclosure at the same wildlife park also displayed clinical signs of coughing at a similar time (Submission 13; Table 1); however, an oral swab, nasal swab, and faecal samples collected on 9 December 2023 were negative for SARS-CoV-2 vRNA.

### 3.5. Submission 15: Dog from England

A previously healthy four-year-old domestic dog (Labrador retriever) was admitted to a veterinary hospital on 6 December 2021 presenting with respiratory, cardiovascular, and gastrointestinal signs which progressed to sudden death. Lethargy, nausea, and vomiting were noted as the initial signs of illness ten days prior to admittance. Three days prior to death, nausea and vomiting continued, with development of a progressive cough resulting in respiratory distress. Clinical assessment also reported cardiac arrhythmia with abdominal distension. The two owners who resided in the same household as the dog had both tested positive for SARS-CoV-2 and their isolation had ended on 3 December 2021. A nasopharyngeal swab was taken from the dog for SARS-CoV-2 RRT-PCR testing on the day of admittance and was identified as positive by PCR by a private laboratory. Extracted RNA from the original nasopharyngeal swab sample was submitted to APHA for SARS-CoV-2 confirmation along with the carcass for post-mortem examination. At necropsy, an enlarged, globoid, heart with a dilated right ventricular wall was noted, and subsequent histopathological findings revealed an extensive, chronic, myocardial fibrosis of the right ventricular wall with focal fibrinoid vasculitis, and as such cardiac failure was attributed as the cause of death. Additionally, there was a focal, infarctive, coagulative necrosis in the kidney identified on histopathology, likely due to thromboembolism related to the perturbed haemodynamic from cardiac failure. Immunohistochemical staining for SARS-CoV-2 specific nucleoprotein was negative in all tissues examined, including the brain, eye, heart, lung, digestive tract, and urinary system.

Nasal and throat swabs were taken post-mortem along with BALF, blood, and multiple tissue samples for RNA extraction and SARS-CoV-2 specific RRT-PCR testing. SARS-CoV-2 vRNA was detected from the original pharyngeal sample (Cq 31.46) collected on 6 December 2021 along with the nasal swab (Cq 32.75), throat swab (Cq 34.75), BALF (Cq 30.50), blood collected from the heart (Cq 33.44), and brain tissue (Cq 31.71) (Table 1) collected seven days later at post-mortem examination. Additional tissue and swab samples collected at post-mortem (including rectal swab, conjunctival swab, preputial swab, abdominal fluid and sections of spleen, kidney, lung, heart, trachea, oesophagus, duodenum, stomach, rectum, pancreas, Ileum, liver, lymph node, colon, thyroid, salivary gland) were negative for SARS-CoV-2 vRNA. Virus isolation was successful from the BALF sample but not from the swabs or brain tissue.

### 3.6. Genome Sequencing and Phylogenetic Analysis

Whole genome sequence data were generated from the positive domestic cat (Submission 2) sample by the original investigators [27] (EPI_ISL_536400; Figure 1) and were characterised as an early SARS-CoV-2 lineage (Pango Lineage B.1.1.142). SARS-CoV-2 genome sequence was obtained from the three dogs (Submissions 7, 9, and 15; Table 1) and three tigers (Submissions 10–12; Table 1) positive for SARS-CoV-2 vRNA and consensus sequences deposited in GISAID EpiCoV under accession numbers: EPI_ISL_18943724–EPI_ISL_18943729 and GenBank accession numbers: PP515674–PP515679 (Table 2). The reference guided assembly files have been deposited in NCBI SRA under experiment accession number PRJNA1092720 and mapping statistics were provided (Appendix A). All sequences were characterised as the Delta VOC (B.1.617.2) with designated sublineages (Table 2). SARS-CoV-2 sequences from the known COVID-19 positive human contacts to these animal cases were unavailable; therefore, the genome sequences were compared with the 15.8 million genome sequences available in the EpiCoV database (GISAID, accessed on 6 July 2023) with the most closely related human sequence identified (Table 2). Nucleotide sequences from each animal case were compared with their respective closest human sequence and the percentage sequence similarity for each comparison given (Table 2) based on sequence coverage available after gaps in the data were removed. Amino acid sequence comparisons of the spike protein from all six animal cases and their respective closest sequence from humans showed no sequence changes at the consensus level. Phylogenetic analyses (Figure 1) show the SARS-CoV-2 positive animal cases clustering with the early B.1 lineage (positive domestic cat) and lineages of the Delta variant (positive domestic dogs and captive Amur tigers).

## 4. Discussion

SARS-CoV-2 infection in animals has been reported since the start of the COVID-19 pandemic. Most notable have been the detection and transmission of SARS-CoV-2 in farmed mink [20,21], white-tailed-deer [22,23,24,25], and companion animals [18,39,40,41] where close interactions with SARS-CoV-2 infected humans have occurred. Among companion animals; cats, ferrets, and hamsters are described to be highly susceptible and most at risk of SARS-CoV-2 infection [18,42]. The first cases of SARS-CoV-2 infection in domestic cats were reported in Hong Kong and New York in March 2020 [43,44] and numerous cases have been reported globally [17], mostly associated with reverse-zoonotic transmission. The first human-to-domestic cat SARS-CoV-2 transmission in the UK in 2020 resulted in mild clinical disease [27] and this is reflected in the majority of cats infected with SARS-CoV-2. While the majority of SARS-CoV-2 infected cats developed subclinical infection or mild disease, limited cases have manifested with severe disease outcomes [45,46,47]. The expression of angiotensin-converting enzyme 2 (ACE2) receptor in tissues examined from cats also indicates that cats are at risk for reverse-zoonotic infection with SARS-CoV-2 [42]. Cases where severe disease in cats has been reported were often associated with other underlying conditions such as feline hypertrophic cardiomyopathy or *Mycoplasma felis* infection where SARS-CoV-2 was an incidental finding rather than the principle cause of death [48,49]. In a serological study, the identification of concomitant infections with other pathogens such as *Toxoplasma gondii*, *Leishmania infantum*, feline leukaemia virus (FeLV), and feline immunodeficiency virus (FIV) in stray cats in Spain highlighted potential increased susceptibility to SARS-CoV-2 infection [50] and exposure to both SARS-CoV-2 and feline coronaviruses have been reported in the UAE [51]. While transmission among cats has been documented experimentally [52,53], domestic cats are not considered potential reservoir hosts for SARS-CoV-2 or to play a significant role in spillback to humans [18].

Two of the three positive dog cases described in this report (Submissions 7 and 15) were either euthanised for clinical welfare or presented as sudden death. For Submission 7, the dog was presented with underlying conditions and although SARS-CoV-2 infection was not likely to be the sole cause of death it cannot be discounted as a contributing factor. As for Submission 15, although vRNA was detected in the brain, respiratory tract, and blood, the absence of viral antigen colocalisation with the diseased heart, and considering the chronicity of the heart lesion, make it difficult to confirm or exclude the potential role of SARS-CoV-2 in disease progression and death of this dog. Dogs have been described as a species that are less susceptible to productive SARS-CoV-2 infection compared to domestic cats [54] with limited angiotensin-converting enzyme 2 (ACE2) receptor labelling in dogs [42]. In these case descriptions, more dogs compared to cats were confirmed positive for SARS-CoV-2 despite comparatively less submissions. This trend has been seen globally since 2020 with 246 dogs (*Canis lupus familiaris*) and 198 cats (*Felis catus*) naturally infected with SARS-CoV-2 reported to the WOAH to date [55] and reported in other case descriptions [56]. However, the total number of each species tested globally remains unclear. Case reports of SARS-CoV-2 serosurveys conducted in dogs and cats are variable with some reporting high levels [57,58,59] and others reporting low prevalence or a complete lack of evidence [60,61]. Higher seroprevalence in cats compared to dogs has been described [62]. However, it has been reported that some cats infected with feline coronavirus can develop cross-reactive antibodies to SARS-CoV-2 that could be contributing to the SARS-CoV-2 seroprevalence seen in cats [63,64]. Without underlying health conditons or co-infections, productive SARS-CoV-2 infection may be subclinical in many species; therefore, animal cases may be under-reported. It is also noted that the regulatory requirement for reporting SARS-CoV-2 infections in animals was established in February 2021; therefore, positive cases of domestic cats and dogs that occurred before this time in GB were not required to be reported with onward supply of material to APHA for confirmatory testing, and thus not captured in this report. SARS-CoV-2 infection in dogs has been described globally with cases related to close human contact [65,66,67,68,69]. A range of disease severity has been seen in dogs naturally infected with SARS-CoV-2 from subclinical [41] to mild respiratory and digestive clinical signs [69]. It has been proposed that the susceptibility of dogs to productive SARS-CoV-2 infection may be influenced by co-morbidities [70] and severe disease resulting in death has been described in dogs where other underlying conditions were present, similar to that of the dog described in this report (Submission 7).

This is the first case report of SARS-CoV-2 infection in tigers in GB; however, tigers (*Panthera tigris*) naturally infected with the Delta variant have been detected in the United States of America [71,72], Argentina, Denmark, Indonesia, and Sweden and reported to the WOAH [17,55]. The three tigers housed in the same enclosure confirmed positive in GB as described in this report may have been exposed to the same source of infection, likely contact with an asymptomatic zoo worker; however, intraspecies or indirect transmission cannot be excluded. The only known worker that was confirmed to be SARS-CoV-2 positive was identified three months prior to the detection in the tigers. However, it cannot be assumed this individual was not the original source of their infection as the incubation period of SARS-CoV-2 in tigers is unknown. SARS-CoV-2 has been detected in other large felids in zoos and wildlife parks including other *Panthera* species such as lions and leopards [55,71,72], further demonstrating the wide host range.

The detection of the Delta VOC in dogs and tigers was between July and December 2021 when the Delta variant was dominant in the human population in England [12]. While these positive animal detections had known or likely contacts with COVID-19 infected owners or workers, it was not possible to obtain sequence from the infected individuals to compare for specific host genetic adaptation markers. The SARS-CoV-2 spike protein sequences from the dog and tiger cases in the UK compared with their respective closest match from humans did not identify any sequence changes and investigations of other cases where sequence was available from infected animals and their human contacts revealed high nucleotide sequence similarities up to 99.9% [73]. Descriptions of animal SARS-CoV-2 cases support reverse zoonosis events where the same variant circulating in the human population mirrors variants found in animals during the same period. A review of animal derived SARS-CoV-2 virus sequences identified unique amino acids in the receptor binding domain of the spike protein [74] but understanding whether these mutations contribute to a higher fitness in the animal host remains unknown. Sustained SARS-CoV-2 transmission in animals introduces the emergence of novel mutations that increase the risk for altered viral fitness, host range expansion, and acquired immunity evasion in humans. An example of this has been seen with the extensive SARS-CoV-2 transmission in white-tailed deer in the USA where Alpha and Gamma VOCs have been detected long after their circulation in the human population diminished [24,25].

The majority of SARS-CoV-2 animal cases reported globally are of Delta or earlier VOCs and there are relatively few reports of animal infections with the Omicron variant and its sublineages [75,76,77]. In the UK, the Omicron variant (BA.1) became dominant in the human population in December 2021 and since then there have only been two submissions to the NRL for SARS-CoV-2 in animals, both confirmed negative for SARS-CoV-2 (to date, January 2024). Several contributing factors may be involved in the lack of Omicron variant detections in animals. Epidemiological and experimental findings have shown that although Omicron and its sublineages have enhanced transmissibility properties, Omicron infection causes less severe disease than Delta in multiple species [78]. A serological field study observed a reduced number of seropositive cats and dogs during the early phases of Omicron circulation in humans [79] and Omicron variant BA.1.1 demonstrated lower pathogenicity compared to earlier variants in a feline experimental model [80]. In hamsters, experimental infections with an Omicron variant showed similar clinical disease compared to Delta variant infection [81]; however, these reports vary depending on the Omicron sublineage [82] and how this reflects natural infection and reverse zoonotic transmission is unclear. A reduced number of Omicron variant infection detections in animals may also be a consequence of increased vaccination uptake in the human population, thereby reducing human infection and/or reduced human surveillance in the UK; therefore, pet owners with asymptomatic and unconfirmed SARS-CoV-2 infection may be unknowingly transmitting SARS-CoV-2 to their pets. The Omicron variant may have become less efficient at transmitting from humans to certain animal species or if clinical outcomes in animals are mild/non-clinical following infection with the Omicron variant then veterinary investigation may not be required.

Despite the limited reports of SARS-CoV-2 Omicron variant infections in animals, continued surveillance in animals will further our understanding on susceptibility and viral pathogenesis in different animal species. Continued circulation in the human population and the emergence of further VOCs may increase the host range or susceptibility of different species and escape from pre-existing immunity upon reinfection. Such work will significantly contribute to ongoing and future risk assessments of SARS-CoV-2 transmission in different species and the threat of zoonotic transmission.

## Figures and Tables

**Figure 1 viruses-16-00617-f001:**
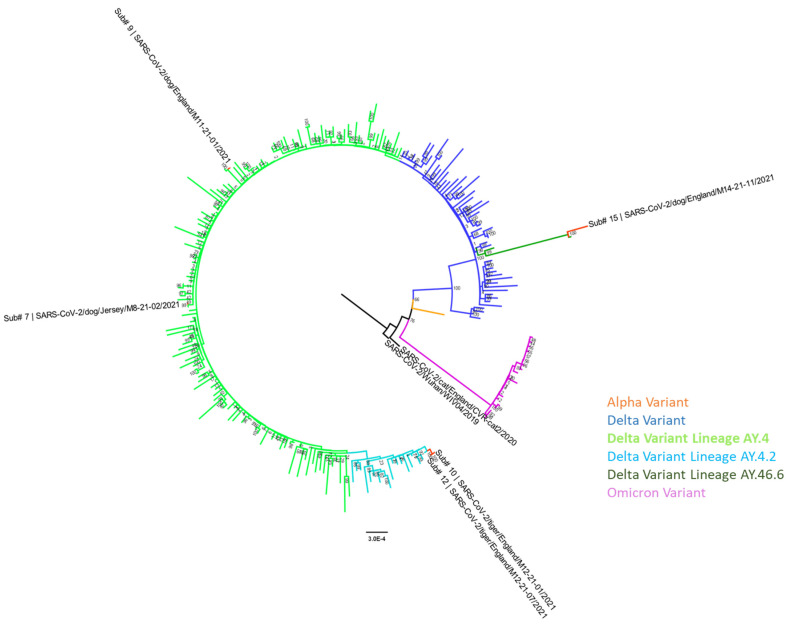
Maximum likelihood phylogenetic analysis of the SARS-CoV-2 genome sequences from positive animal cases in Great Britain and Jersey with representative human sequences obtained from GISAID from May 2020 to December 2021. Sequence from Submission 11 omitted in the analyses due to partial sequence obtained. The maximum likelihood phylogenetic analysis was inferred using the General Time Reversible (GTR) substitution model with empirical base frequencies (+F) and four categories of rate variation (+R4) with a phylogeny test of 1000 ultrafast bootstrap replicates.

**Table 2 viruses-16-00617-t002:** Delta variants and PANGO sublineages of the SARS-CoV-2 genomes sequenced from the dog and tiger animal cases and comparisons with the closest human genome sequence identified on EpiCoV (GISAID).

Submission #	Sequence Name	Sample Type	Sample Collection Date	Accession Numbers GISAID; GenBank	Delta Variant PANGO Sublineage	Most Closely Related Human Sequence	Percentage Nucleotide Similarity (Genome Coverage)
7	SARS-CoV-2/dog/Jersey/M8-21-02/2021	Conjunctival and Oropharyngeal swab pooled	9 July 2021	EPI_ISL_18943724; PP515674	AY.4	SARS-CoV-2/England/PHEC-3461AE/2021 collection date: 2021 EPI_ISL_3572506	99.95% (91%)
9	SARS-CoV-2/dog/England/M11-21-01/2021	Pharyngeal swab	29 October 2021	EPI_ISL_18943725; PP515675	AY.4	SARS-CoV-2/Scotland/QEUH-29B97E4/2021 collection date: 08-11-2021 EPI_ISL_6433530	100% (90%)
10	SARS-CoV-2/tiger/England/M12-21-01/2021	Sputum swab	3 December 2021	EPI_ISL_18943726; PP515676	AY.4.2	SARS-CoV-2/England/QEUH-27CC601/2021 collection date: 19-10-2021 EPI_ISL_5529494	99.91–99.93% (90–93%)
11	SARS-CoV-2/tiger/England/M12-21-05/2021	Nasal swab	9 December 2021	EPI_ISL_18943727; PP515677	AY.4.2		
12	SARS-CoV-2/tiger/England/M12-21-07/2021	Nasal swab	9 December 2021	EPI_ISL_18943728; PP515678	AY.4.2		
15	SARS-CoV-2/dog/England/M14-21-11/2021	BALF	13 December 2021	EPI_ISL_18943729; PP515679	AY.46.6	SARS-CoV-2/England/MILK-2E31E35/2021 collection date: 11-12-2021 EPI_ISL_7815416	99.99% (98%)

## Data Availability

Genetic data from this study has been made available through the Global Initiative on Sharing Avian Influenza Data (GISAID) EpiCoV platform (https://www.GISAID.org) under accession numbers: EPI_ISL_18943724–EPI_ISL_18943729 and GenBank accession numbers: PP515674–PP515679. The reference guided assembly files have been deposited in NCBI SRA under experiment accession number PRJNA1092720.

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
