# Peer review of "Detection of SARS-CoV-2 Delta Variant (B.1.617.2) in Domestic Dogs and Zoo Tigers in England and Jersey during 2021"

_viruses, 2024, doi:10.3390/v16040617_

Round 1
Reviewer 1 Report
Comments and Suggestions for Authors
The authors must provide much more detailed information on sequencing and the mapping of reads to the reference SARS-CoV-2 sequence. The coverage, standard deviation, quality scores of each base of the assemblies (fastq scores of each assembled base) have to be clearly described in the appropriate Material and Methods and Results sections. The sequencing raw data has to be deposited in NCBI SRA. How the exported consensus sequences of each whole genome sequence were obtained? 50% majoriry rule, 60% quality? This must be especified. The consensus sequence of each sample must be deposited on GenBank and the accession numbers available in the manuscript. The boosttrap values of branch clusters of the maximum likelihood phylogenies must be presented. Also, the choice of the model used for the maximum likelihood inference must be presenteted. Was modeltest used? How were parameter models inferred?
Author Response
The authors must provide much more detailed information on sequencing and the mapping of reads to the reference SARS-CoV-2 sequence.
Thank you for your review. We acknowledge that it is necessary to provide more detailed information on the sequencing and mapping of reads, please see below answers to the specific points raised:
- The coverage, standard deviation, quality scores of each base of the assemblies (fastq scores of each assembled base) have to be clearly described in the appropriate Material and Methods and Results sections.
The mapping statistics has been generated using samtools and added to the Materials and Methods sections with the results provided in a new supplementary file, Supplementary Table 1. We acknowledge that it is necessary to provide more detailed information on the sequencing and mapping of reads and the appropriate statistics for this manuscript include the number of reads in the assembly, the percentage coverage of positions, the mean depth of coverage across the genome, the mean base quality score and the mean mapping quality of the selected reads.
- The sequencing raw data has to be deposited in NCBI SRA.
We are in the process of further analysing the raw sequencing data beyond the scope of SARS-CoV-2 investigation that is not relevant to this manuscript, once this is completed we will deposit in NCBI SRA. Currently we are not permitted to publicly share this data from the diagnostic animal cases. However, to share the information on the relevant reads obtained for these sequences the reference guided assembly bam file has been deposited in NCBI SRA under experiment reference number PRJNA1092720. This has been added to the revised manuscript.
- How the exported consensus sequences of each whole genome sequence were obtained? 50% majoriry rule, 60% quality? This must be especified.
The final consensus was generated using iVar with a minimum depth of one sequence with a 50% minimum frequency rule for degenerate base calling. The quality threshold was a Phred score of 20. These details have been added to the methods.
- The consensus sequence of each sample must be deposited on GenBank and the accession numbers available in the manuscript.
The consensus sequences have already been deposited on GISAID which remains the primary repository for SARS-CoV-2 genomes globally. However, as requested, the consensus sequences have also been submitted to GenBank and the accession numbers provided in Table 2 along with the GISAID accession numbers.
- The boosttrap values of branch clusters of the maximum likelihood phylogenies must be presented.
An updated phylogenetic tree has been provided (Figure 1) that includes bootstrap values.
- Also, the choice of the model used for the maximum likelihood inference must be presenteted. Was modeltest used? How were parameter models inferred?
A best-fit model was applied using ModelFinder. The maximum likelihood phylogenetic analysis was inferred using the General Time Reversible (GTR) substitution model with empirical base frequencies (+F) and four categories of rate variation (+R4) with a phylogeny test of 1000 ultrafast bootstrap replicates. This information has been added to the methods and results sections with additional references cited to support their use.
Reviewer 2 Report
Comments and Suggestions for Authors
The manuscript documented the occurrence of SARS-CoV-2 Delta Variant (B.1.617.2) in domestic dogs and zoo tigers.
The title of the manuscript seems adequate.
While the subject is not new, its justification is clear and of value for diagnosticians and researchers.
The design of the study is adequate and its conclusions and interpretations valid. The figures and their captions or legends as wells as the tables and titles are adequate.
For clarity, it is recommended to the authors to revise the following:
Lines 114, 115: “Samples used for histopathological and in situ detection of SARS-CoV-2 were fixed in 10% buffered formalin and routinely processed as described [26].”
The sentence should read: “…fixed in neutral-buffered, 10% formalin fixative solution…”
Author Response
Lines 114, 115: “Samples used for histopathological and in situ detection of SARS-CoV-2 were fixed in 10% buffered formalin and routinely processed as described [26].
The sentence should read: “…fixed in neutral-buffered, 10% formalin fixative solution…”
Thank you for your review, the sentence has been amended as requested.